# Bacteriostatic Effects of Yujin Powder and Its Components on Clinical Isolation of Multidrug-Resistant Avian Pathogenic *Escherichia coli*

**DOI:** 10.3390/vetsci10050328

**Published:** 2023-05-04

**Authors:** Jinwu Meng, Jinli Wang, Jinyue Zhu, Siya Li, Tianxin Qiu, Weiran Wang, Jinxue Ding, Wenjia Wang, Jiaguo Liu

**Affiliations:** 1MOE Joint International Research Laboratory of Animal Health and Food Safety and Traditional Chinese Veterinary Medicine Research Center, College of Veterinary Medicine, Nanjing Agricultural University, Nanjing 210095, China; 2College of Agriculture, Jinhua Polytechnic, Jinhua 321000, China

**Keywords:** *E. coli*, resistance, Yujin powder, Scutellariae Radix, baicalin, anti-bacterial effect

## Abstract

**Simple Summary:**

Yujin powder is a polyherbal formulation that has been used to treat the large intestine dampness-heat syndrome for a long time. *Escherichia coli* is one of the most common pathogenic bacteria to cause this syndrome in diarrheal chickens, leading to serious economic losses in the poultry industry. The limited effect of antibiotics on antibiotic-resistant *E. coli* makes this bacterium a potential threat to human health. The present study is designed to explore the treatment effects and mechanisms of the Yujin powder and its potential bioactive components on lethal diarrhea caused by clinical isolated multidrug-resistant avian pathogenic *E. coli* and demonstrate the potential bioactive component of YJP. A multidrug-resistant *E. coli* was isolated and identified from a clinical diarrheal chick. Yujin powder and its components could directly inhibit the growth of this strain at high concentrations in vitro, and presents obvious anti-bacterial effects by reducing the bacterial loads, the release of endotoxin, and inflammation in vivo, which was much more effective than the resistant antibiotic ciprofloxacin. Based on our results, Yujin powder and its components, *Scutellariae Radix* and Baicalin, could be used as novel treatment against the diarrhea caused by this isolated multidrug-resistant *E. coli*.

**Abstract:**

*Escherichia coli* is one of the most common pathogenic bacteria in diarrheal chickens, leading to serious economic losses in the poultry industry. The limited effect of antibiotics on antibiotic-resistant *E. coli* makes this bacterium a potential threat to human health. Yujin powder (YJP) has been reported as an agent that releases the symptoms caused by *E. coli* for a long time. The objective of this study is to investigate the effect of Yujin powder (YJP) and its components, Scutellariae Radix (SR) and Baicalin (Bac), anti-against multi-drug-resistant *E. coli* in vitro and in vivo. A multi-drug-resistant bacteria was isolated and identified from a clinical diarrheal chick. Then, the anti-bacterial effects of drugs were assessed in vitro and in vivo by analyzing the bacteria loads of organs, the levels of endotoxin, TNF-α, IL-1β, and IL-6 of the serum. Results found that the pathogenic *E. coli* was resistant to 19 tested antibiotics. YJP, SR, and Bac could directly inhibit the growth of this strain at high concentrations in vitro, and presents obvious anti-bacterial effects by reducing the bacterial loads, the release of endotoxin, and inflammation in vivo, which was much more effective than the resistant antibiotic ciprofloxacin. This study demonstrates that those natural medicines have the potential to be used as novel treatments to treat the disease caused by this isolated MDREC strain.

## 1. Introduction

Avian pathogenic *Escherichia coli* (APEC) causes bacterial infections through the digestive and respiratory tract in poultry, leading to colibacillosis [1]. Nowadays, colibacillosis caused by APEC is one of the major causes of high mortality and morbidity, which causes great economic losses in the poultry industry [2,3]. Antibiotics were used to promote the growth of animals or treat diarrhea caused by pathogenic bacteria in the chicken industry [4,5]. However, the abuse and long-term use of antibiotics lead to the production of drug-resistant strains, which makes antibiotics ineffective, brings food safety to human beings, and also formats a serious threat to human health [6,7]. Nowadays, more and more reports found that multiple strains of APEC isolated from farms with several antibiotic-resistant genes are resistant to multiple antibiotics, even APEC strains isolated from one farm are all resistant to some commonly used antibiotics in clinical practice [8,9,10]. Therefore, it is urgent to develop safe, efficacious agents with limited harmful effects for the treatment of multi-drug-resistant *E. coli* (MDREC) infected diarrhea.

Clinical practices in traditional Chinese medicine (TCM) and traditional Chinese veterinary medicine (TCVM) showed that the use of TCMs exerted great effects to treat diarrheal diseases under the guidance of TCM theory, especially the use of formulas containing several herbs [11,12]. For instance, Yujin Powder (YJP), one classical prescription composed of various herbs in TCM and TCVM, showed effective therapeutic effects on “Chang Huang”, characterized by fever, diarrhea, and abdominal pain, caused by excessive labor and diet, indigestible food, or contamination of food and drinking water in hot climates [13,14,15]. YJP was first recorded in Yuan Heng’s classical Collection on the Treatment of Equine Diseases with eight herbs, Curcumae Radix, Chebulae Fructus, Scutellariae Radix (SR), Rhei Radix Et Rhizoma (Rhubarb, Rb), Coptids Rhizoma, Phellodendri Chinensis Cortex, Gardenia Fructus, and Paeoniae Radix Alba, based on a Huanglianjiedu decoction [16,17]. According to TCVM theory, chicken colibacillosis belongs to the syndrome of internal heat caused by the accumulation of damp heat. Excessive water and dampness damage the spleen with the decrease of its transport function, leading to symptoms such as diarrhea [13,14]. Coptids Rhizoma, Scutellariae Radix, Phellodendri Chinensis Cortex, and Gardenia Fructus could clear the fire of “Sanjiao” and eliminate damp heat, act as minister herbs in this formula; Curcumae Radix could promote Qi circulation and relieve depression, cool and mix blood, normalize gallbladder to cure jaundice, and act as monarch herb; Paeoniae Radix Alba and Chebulae Fructus could astringe intestines to stop diarrhea, Rhubarb could remove accumulation with purgation, cool the blood, and remove the toxic materials; these three herbs act as adjuvant herbs [18,19,20,21]. Although these herbs have some similar effects, Scutellariae Radix and Rhei et Radix Rhizoma were chosen for further research according to the consumption, the differences among their “flavor and meridian tropism”, and our former research [22,23,24,25]. Nowadays, more and more reports demonstrate that these TCMs are shown to be great treatments and inhibit tissue injury through different pathways according to modern pharmacological research [26,27]. Scutellariae Radix (SR) and its component, baicalin, could decrease the levels of pro-inflammatory cytokines in serum, it also could inhibit the growth of *Staphylococcus saprophyticus* and *Pseudomonas aeruginosa* [24,28,29]. Curcumae Radix could not only directly inhibit bacteria, but also improve the effectiveness of antibiotics on inhibiting the growth of bacteria; so, this herb was not used as the research object of this study [30,31]. Rhubarb demonstrated broad-spectrum antibacterial activity against the Gram-negative bacteria, *E. coli*, and can be used to protect the gut barrier, maintain the intestinal micro-ecological environment, and prevent bacterial translocation as well [32,33]. In these aspects, the concepts of clearing away heat and detoxicating in TCM and TVCM are consistent with that of antibacterial, anti-inflammatory, and diarrhea in modern pharmacological research.

However, the treatment mechanisms of YJP and its potential bioactive components on diarrhea infected by a single MDREC are uncleared. In the present study, we isolate the pathogenic bacteria causing diarrhea in a chicken farm and tested its antibiotic-resistant spectrum. Then the anti-bacterial activities of YJP and its components to this strain were tested and the mechanisms of the effects were demonstrated in vitro and in vivo.

## 2. Materials and Methods

### 2.1. Bacterial Strains

The isolated multi-drug-resistant *E. coli* (MDREC) was obtained from the heart sample collected from a sick chick in a suspected colibacillosis layer farm in Dangtu County, Anhui Province, China. The standard strain of *E. coli* ATCC25922 was presented by a professor of the Department of Basic Veterinary Medicine, Nanjing Agricultural University.

### 2.2. Isolation and Identification of the Pathogen

The heart sample from the sick chickens was inoculated on nutrient broth agar medium (Qingdao Haibo, Qingdao, China) and incubated at 37 °C for 24 h. Then, the bacteria were purified and inoculated on McConkey’s agar medium and eosin methylene blue agar medium (Qingdao Haibo, Qingdao, China). After incubation at 37 °C for 24 h, the isolated bacterium was tentatively identified based on the colony and cell morphology, Gram staining, and biochemical activity detection. Molecular identification of the isolated *E. coli* was performed based on the nucleotide sequence analysis of 16S rRNA (27F: AGAGTTTGATCCTGGCTCAG; 1492R: TACGACTTAACCCCAATCGC). The PCR products were purified and sequenced after being subjected to agarose gel electrophoresis. The sequences were compared with the data using the nucleotide basic local alignment search tool (http://blast.ncbi.nlm.nih.gov/Blast.cgi, accessed on 26 October 2022). Phylogenetic analysis of the 16S rRNA gene was carried out using the MEGA version 7.0 software. The dendrogram was generated by the neighbor-joining method.

### 2.3. Antimicrobial Susceptibility Testing

Nineteen antibiotics were tested by the Kirby–Bauer method on Mueller–Hinton agar according to the guidelines of the Clinical and Laboratory Standards Institute [29]. Interpretive criteria for the inhibition zone diameters were provided by CLSI as well. It was considered as MDREC if the isolates were resistant to more than three classes of antibiotics. *E. coli* ATCC25922 was used as a standard strain to detect the susceptibility of antibiotics disks.

### 2.4. Drug Extracting Methods

The herbs for YJP were purchased from Nanjing Tianyuan Pharmacy (Nanjing, Jiangsu). All these herbs were genuine medicinal materials to ensure the best curative effects and authenticated by prof. Jiaguo Liu in the College of Veterinary Medicine of Nanjing Agriculture University. The composition of YJP was listed in Table 1 [17,34]. The plant names have been checked with “World Flora Online” (www.worldfloraonline.org, accessed on accessed on 17 January 2023). YJP and SR were extracted by the decocting method.

When extracting YJP, all herbs (160 g) except Radix et Rhizoma Rhei (60 g) and Curcumae Radix (30 g), were mixed and immersed in distilled water (2500 mL) for 30 min and then boiled for 30 min; after filtered with 8 layers of gauze, distilled water (1500 mL) was added to the herbs and boiled again; the third extraction is the same as the second extraction; at last, the filtrates from the three times were mixed, filtered by a Buchner funnel, concentrated with a rotary evaporator at 60 °C, and then freeze-dried with a vacuum freeze dryer. Radix et Rhizoma Rhei (60 g) and Curcumae Radix (30 g) were crushed, put into bags, and added in at the last five minutes. SR (30 g) was directly decocted with distilled water (300 mL) as the main method of YJP. When extracting Rb, it (60 g) was soaked in distilled water (600 mL) for 45 min, then decocted in boiling water for 5 min, and repeated three times. The filtrates were treated as YJP. These freeze-dried powders were stored at −20 °C for standby.

### 2.5. The MICs of YJP and Its Components

The minimal inhibit concentrations (MICs) of ciprofloxacin (CIP), YJP, SR, Rb, and Baicalin (Bac) against the MDREC and ATCC25922 were determined by broth microdilution method according to the guideline by the CLSI [35]. The test medium was Mueller–Hinton broth (MHB) and the isolated bacteria were added to the 96-well microtiter plate at an initial concentration of 2 × 10^5^ CFU/mL. Each well contains 50 μL of bacteria and 50 μL of different concentrations of drugs. After incubation at 37 °C for 24 h, the lowest concentration of the drugs without visual growth was recorded as the MIC values of these drugs.

### 2.6. Compliance with Ethical Standards

All animal research and facilities were carried out according to the experimental practices and standards. All experiments comply with the manual of the care and use of laboratory animals published by the US National Institutes of Health. All experimental protocols were approved by the Laboratory Animal Center of Nanjing Agricultural University.

### 2.7. YJP and Its Components against MDREC In Vivo

Institute of Cancer Research (ICR) mice (18~20 g) were purchased from Qing Long Shan animal breeding farm (SCXK (Su) 2017-0001). After 7 d of adaptive feedings, they were divided into six groups, including the blank control (BC) group, the *E. coli* control (EC) group, the CIP group, the YJP group, the SR group, and the Bac group. Each mouse was inoculated with 0.2 mL 4.49 × 10^8^ CFU/mL bacteria by intraperitoneal injection according to the toxicity test results of isolated bacteria (Appendix A), except the BC group with normal saline. After one hour, the treatment groups were treated with YJP (3.75 g/kg) and SR (1 g/kg) by intragastric administration, Bac (0.0375 g/kg) by hypodermic injection near the back, and CIP (0.02 g/kg) by intramuscular injection in the thighs, respectively. The drugs were given once a day for 3 d continuously. The BC group and EC group were treated with the same amount of normal saline at the same time. The clinical symptoms and mortality of mice were observed and recorded for 7 d continuously.

### 2.8. Histopathological Evaluation

These mice were anesthetized with CO_2_ and the spleen tissues were obtained in each group at 18 and 72 h after injecting the isolated bacteria liquid. Sections (5-mm thickness) of fully fixed spleens were obtained and stained with hematoxylin and eosin, and then examined under a light microscope (Olympus Corporation, Tokyo, Japan).

### 2.9. Bacterial Loads in Organs

The heart, liver, and spleen were ground with 1 mL of normal saline by the homogenizer. The upper suspensions were diluted 10 times and inoculated to MacConkey agar plates with 20 μg/mL ampicillin. After incubation at 37 °C for 18 h, the numbers of viable bacteria in 1 g tissue were measured and calculated.

### 2.10. Endotoxin and Indexes of Inflammatory

The blood samples at 18 and 72 h after injecting the isolated bacteria were centrifuged to collect serum after coagulation at 37 °C and stood for 30 min. The levels of endotoxin, TNF-α, IL-1β, and IL-6 in the serum were determined according to the instructions of ELISA kits (Nanjing Aoqing, China).

### 2.11. Statistical Analysis

All data were analyzed by IBM SPSS statistics 19.0 software (IBM Corporation, Armonk, NY, USA). The statistical results were expressed as mean ± standard deviation. The difference among the mean values of experimental groups was analyzed by one-way ANOVA and Duncan’s multiple range test. The survival rate was tested by the chi-square test. *p*-values < 0.05 were considered statistically significant.

## 3. Results

### 3.1. The Colony Morphology of the Isolate

A large number of colonies with consistent morphology indicated that the diarrhea was caused by a single pathogen infection. On nutrient agar medium, the colonies are different in size but consistent in colony morphology, about 1–3 mm in diameter, moist and smooth on the surface, and neat in edges (Figure 1A). The diameters of colonies on McConkey’s medium are about 2 mm, the colonies are smooth, flat and moist, neat round, with neat edges, and a single colony is light red (Figure 1B). On eosin-methylene blue agar medium, the colonies’ diameters are about 1 mm, the purple-black colonies are round and convex, with smooth surfaces and typical metallic luster (Figure 1C). The typical dominant single colonies on the identification medium were selected for purification and culture for subsequent experiments.

### 3.2. Biochemical Identification of the Isolate

The results of the biochemical identification of the isolate are shown in Table 2. On trisaccharide iron medium (TSI), the isolate can produce acid and gas on both the slope and the bottom layer, but cannot produce H_2_S. Indole can be produced with the forming of red ring. Glucose, lactose, maltose, mannitol, and sucrose can be fermented with the appearance of acid and gas. It can ferment honey disaccharide, cannot ferment side marigold alcohol and inositol, cannot utilize citrate, and cannot decompose urea. The VP test is negative, and the MR is positive. The lysine decarboxylase test is positive, and the phenylalanine deaminase is negative. It is athletic and can grow along the puncture line in semi-solid agar.

### 3.3. Microscopic Identification and PCR Identification

The Gram-negative, short rod-shaped bacteria with blunt ends were seen under the light microscope (Figure 1D). Then the molecular method was applied to identify the isolate. The 16S rRNA gene was amplified and sequenced. The isolate was closely similar to *E. coli* with the highest scoring 99% from the BLAST program and the phylogenetic tree analysis (Figure 1F). Combined with the results of other identification tests mentioned above, it can be proved that the isolated pathogen is *E. coli*.

### 3.4. Antimicrobial Susceptibility Testing

The diameters of the inhibition zone and the antimicrobial susceptibility results were shown in Table 3. The isolate is resistant to all the tested 19 antibiotics, indicating that the pathogen is MDREC.

### 3.5. Antimicrobial Activity of CIP, YJP, and Its Components

We detected the MICs of YJP, Rb, SR, and Bac to assess the antibacterial effect of YJP and its components on the isolated MDREC. YJP and its components demonstrated antibacterial activity against MDREC and ATCC25922 in vitro, with MIC of YJP values 250.0 mg/mL, Rb 250.0 mg/mL, SR 62.5 mg/mL, and Bac 2.0 mg/mL. The MIC of CIP for MDREC was 8.0 × 10^−2^ mg/mL, while for ATCC25922 was 1.0 × 10^−5^ mg/mL, indicating the severe resistance of the isolate to CIP (Table 4). After co-culture with 1/2 MIC concentration of Rb and SR for 24 h, the number of viable bacteria was significantly less than that in the blank group (Appendix A).

### 3.6. The Survival Rate of Each Group of Infected Mice

The survival rate of mice in each group is shown in Table 5. Survival rate (%) = number of survivors/total number of effective samples × 100%. There is no death case in the BC group, and the survival rate in the EC group is significantly lower than that in the BC group. The survival rate of the YJP group (66.7%), the SR group (66.7%), and the Bac (73.3%) group were significantly higher than that of the EC group (26.7%) (*p* < 0.05). In addition, the survival rate of the Bac group was significantly higher than that of the CIP group.

### 3.7. Histopathological Changes of the Spleen of Mice

The sections of spleens in each group were evaluated microscopically using H + E staining, and the results were shown in Figure 2. At 18 h, the boundary between red pulp and white pulp in the BC group was clear, and the structure of the splenic corpuscle was complete. In the CIP group and EC Group, the boundary between red pulp and white pulp of spleen tissue was blurred, the tissue structure was lost. Increased red blood cells and pink amyloid in the red pulp were observed as well. Many inflammatory cells infiltrated in the EC group. The boundary between red pulp and white pulp in the YJP group was blurred, and that in the SR group and Bac group were clear. There were pink amyloid substances in the red pulp of the three groups, accompanied by a large number of red blood cells and inflammatory cells.

At 72 h, the unique structure of normal spleen tissue disappeared, no boundary between red pulp and white pulp was found, a large number of red blood cells and inflammatory cells were found, and pink amyloid diffused in the CIP group and EC Group. In the YJP group, SR group, and Bac group, the boundary between red pulp and white pulp of spleen tissue was clear, and there were still many red cells and inflammatory cells.

### 3.8. Bacteria Loads in Various Organs of Mice

At 18 h, the initial stage of disease, the numbers of colonies in the YJP group, SR group, and Bac group in the liver (Figure 3A), heart (Figure 3B), and spleen (Figure 3C) were not significantly different from that in the EC group (*p* > 0.05). At 72 h, the number of bacteria in the three organs in the YJP group, SR group, and Bac group was significantly lower than that in the EC Group and CIP group (*p* < 0.05), and there was no significant difference among the three TCMs-treated groups (*p* > 0.05). In the YJP group, SR group, and Bac group, which are significantly different from the EC Group, Bac has a more obvious effect in reducing the total number of colonies.

### 3.9. Assessment of Inflammatory Cytokines Levels

The finding of the serum levels of TNF-α (Figure 3D), IL-1β (Figure 3E), and IL-6 (Figure 3F) demonstrated that the YJP, SR, and Bac significantly decreased the inflammation index levels compared with the EC group (*p* < 0.05). Bac could even reduce these three inflammatory factors to the normal level. While, CIP failed to reduce the levels of these three inflammatory factors at 18 h and 72 h, and even slightly increased compared with the EC group with no significance (*p* > 0.05).

### 3.10. The Levels of Bacterial Endotoxin in Serum

As shown in Figure 3G, plasma was significantly affected by the treatment of YJP, SR, and Bac (*p* < 0.05). In YJP, SR, and Bac groups, compared with the BC group, the YJP, SR, and Bac decreased the plasma endotoxin content of mice. In addition, the Bac significantly decreased the plasma endotoxin of mice at 18 and 72 h after inoculating the isolated *E. coli*, compared with the YJP group. While, CIP slightly increased the endotoxin in serum compared with the EC group (*p* > 0.05).

## 4. Discussion

Chicken colibacillosis caused by APEC is one of the important infectious diseases endangering the poultry industry in the world, accounting for the first place of bacterial diseases [2,36]. We received sick chickens from Anhui Dangtu laying hen farm. A large number of chickens on this farm suffered from diarrhea repeatedly, with ineffective use of several antibiotics and a lot of death. In this experiment, the pathogenic bacteria were successfully isolated, identified, and analyzed for drug resistance, and the pathogen was determined to be MDREC.

Efficient alternative treatment must be developed for serious infections caused by multi-drug-resistant bacteria. The current therapeutic strategy by enhancing the dosage of useless antibiotics is problematic due to the development of antibiotic resistance. Antibiotics were always used as growth stimulants, disease prevention, and treatment in animals, which may leave antibiotic residues in foodstuffs such as eggs and meat in the poultry industry, and antibiotic-resistant bacteria appeared at the same time [6,37]. Not only do the residual antibiotics in food produce a series of toxic side effects when they enter the human body, but also the development of drug-resistant bacteria poses challenges to the treatment of human diseases [38,39]. In this study, the isolated pathogenic *E. coli* was resistant to all tested bacteria, and the MIC of CIP to this strain was 8000 times the amount of that to ATCC 25922, indicating the severe antibiotic resistance caused the ineffectiveness of several antibiotics in the treatment. 

Recent studies have reported that several Chinese herbal compounds exert potent activities against pathogenic bacteria [40,41]. Previously, it was confirmed that YJP could effectively treat diarrhea caused by *E. coli* [13,14,42]. However, whether YJP exerted its effects and which was the bio-effective component remained unknown. To further explore the potential mechanisms, the MICs of YJP, SR, Rb, and Bac were investigated. MIC was typically used to reveal the anti-bacterial effect of drugs [29,38]. Results of MICs showed that YJP, SR Rb, and Bac could directly suppress the bacteria growth at high concentrations in vitro.

As an important indicator of the drug protection effect, the survival rate of experimental animals can directly indicate the therapeutic significance of drugs [43,44,45]. The treatment of YJP, SR, and Bac significantly increased the number of surviving mice. The results showed that YJP, SR, and Bac could effectively reduce the mortality of the diseased mice, showing a good anti-infection effect, among which Bac had the best effect. The pathogenic mechanism of pathogenic *E. coli* is a multi-step mechanism, including the colonization of mucosal sites, escape from host defense, bacterial proliferation, and damage to host tissue and cells [46]. At the beginning of the disease (18 h), bacterial proliferation and colonization were dominant, there was no difference in the bacteria loads between all groups in all tissues. At the late stage (72 h), the total number of colonies in the YJP group, SR group, and Bac group was significantly lower than that in the EC group and CIP group. It can be seen that the total number of colonies in various organs could be effectively reduced by YJP, SR, and Bac. In addition, the groups with high survival rates also have fewer bacterial loads in the organs.

Endotoxin exists in the cell wall of Gram-negative bacteria and can be released when the cell wall disintegrates after the death of the bacteria [47]. When *E. coli* invade the body, it will release endotoxin [48]. At 18 and 72 h after inoculation, the serum endotoxin of the YJP, SR, and Bac groups was reduced to the normal level. Bac had a better effect on endotoxin reduction than YJP and SR at 72 h. After CIP treatment, the content of endotoxin slightly increased compared with that of the EC group (*p* > 0.05), corresponding to the conclusion that antibiotic-treated animals may show higher endotoxin levels than untreated animals [49,50]. All TCMs can effectively eliminate endotoxin in the body and improve the survival rate. When coliform infection is clinically treated with antibiotics, it may increase the release of endotoxin, leading to further deterioration of the disease [51].

The release of endotoxin can activate the monocyte-macrophage system and induce secretion of various mediators, including TNF-α, Cytokines such as IL-1β and IL-6, which are most closely related to the pathological response of the body [52,53]. The data from this study showed that the MDREC infection can cause severe inflammatory reactions in the body as evidenced by increased plasma TNF-α, IL-1β, and IL-6 levels and spleen pathological changes of inflammatory reaction. At 18 and 72 h after inoculation, the inflammatory factors can be significantly reduced with the treatment of YJP, SR, and Bac, corresponding to the results of spleen section staining. Among them, Bac-reduced IL-1β and IL-6 was more effective than YJP and SR. However, these effects were not observed in the CIP group, indicating the failure of CIP treatment because of antibiotic resistance [54]. The results were in accordance with the conclusion that reducing the levels of inflammatory factors is one of the mechanisms by which drugs play a therapeutic role in infectious diseases [55].

Our results indicated that Bac may be the main bioactive component of SR, which may be an important active drug of YJP. In our study, the effect of Bac is best, followed by SR, and YJP. However, in clinical practice, there are few cases of single bacterial infection, and it is generally multiple bacterial mixed infections. Therefore, the use of YJP always has a better effect than a single drug in clinical practice, which is always different from the results of the laboratory [56,57]. If diarrhea caused by a single bacterial infection is confirmed in clinical practice, the use of baicalin may have a better therapeutic effect and less cost than the use of formula.

## 5. Conclusions

In conclusion, the pathogenic MDR *E. coli* causing chicken diarrhea was successfully isolated and identified, and YJP and its components could inhibit this strain in vitro and in vivo. Moreover, the antibacterial effects of YJP, SR, and Bac were achieved by reducing the bacterial loads, the release of endotoxin, and inflammation. YJP, SR, and Bac could be potent antibiotic substitutes to treat colibacillosis caused by this strain.

## Figures and Tables

**Figure 1 vetsci-10-00328-f001:**
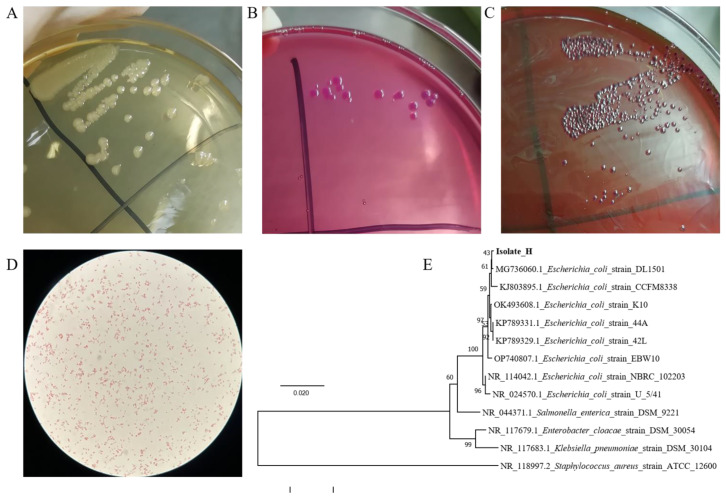
The characteristics of the isolate. (**A**–**C**). The colony morphology of the isolate from the heart on various media. (**A**). Nutrient Agar medium; (**B**). MacConkey agar medium (**C**). Eosin-methylene blue agar medium; (**D**). Morphology of isolates light microscope (10 × 100). The Gram-negative, short rod-shaped bacteria with blunt ends were seen under a light microscope. (**E**). Phylogenetic tree analysis of isolate and other bacteria. Phylogenetic tree analysis of this strain was based on 16S rRNA sequences (1433 bp fragments).

**Figure 2 vetsci-10-00328-f002:**
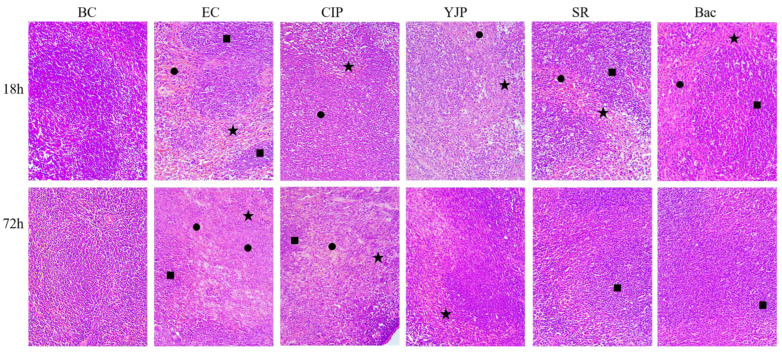
Effects of CIP, YJP and its components on splenic histopathological changes of mice with *E. coli* infection (H&E staining, ×200, n = 6). BC, Blank Control; EC, *E. coli* control; YJP, Yujin Powder; SR, Scutellariae Radix; Bac, Baicalin; CIP, ciprofloxacin. ★, blood cells; ●, pink amyloid; ■, inflammatory cell infiltration.

**Figure 3 vetsci-10-00328-f003:**
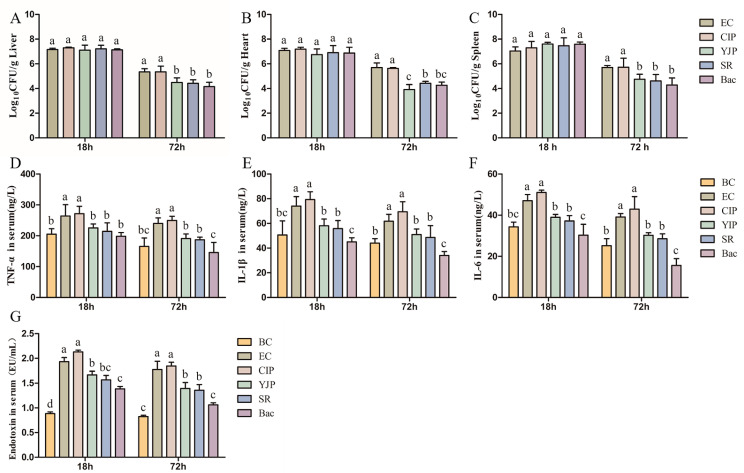
Treatment effects of CIP, YJP and its components on the mice with *Escherichia coli* infection (*n* = 6). (**A**–**C**). The logarithm of the number of bacteria in different organs ((**A**) Liver; (**B**) Heart; (**C**) Spleen). (**D**–**F**). The contents of inflammatory factors in the serum of different treated groups. ((**D**) TNF-α; (**E**) IL-1β; (**F**) IL-6). (**G**). The contents of endotoxin in serum of different treated groups. BC, Blank Control; EC, *E. coli* control; YJP, Yujin Powder; SR, Scutellariae Radix; Bac, Baicalin; CIP, ciprofloxacin. The data was represented by the mean ± standard deviation. The letter “a” indicates the maximum value, “a”, “b”, “c” and “d” decrease in turn. The same lowercase letter indicates no significant differences between groups, while completely different lowercase letters show significant differences with a *p* < 0.05.

**Table 1 vetsci-10-00328-t001:** Herbs of Yujin Powder (YJP).

Traditional Chinese Medicine	Medicinal Plant	Medicinal Parts	Voucher Specimen Number	Local Name	Amount (g)	Origin (China)	Effects [17,34]
Curcumae radix	*Curcuma wenyujin* Y.H.*Chen et* C.*Ling*	Radix	NJAU-CVM-2018001	Yu Jin	30	Zhejiang province	Promote qi circulation and relieve depression, cool and mix blood, and normalize gallbladder to cure jaundice
Chebulae fructus	*Terminalia chebula Retz*	Fructus	NJAU-CVM-2018002	He Zi	15	Guangzhou province	Restraining intestine to stop diarrhea, astringe the lung to relieve asthma
Scutellariae radix	*Scutellaria baicalensis Georgi*	Radix	NJAU-CVM-2018003	Huang Qin	30	Liaoning province	Clear heat and dry damp, clear away heat and toxic material
Rhei radix et rhizoma	*Rheum palmatum* L.	Radix et Rhizoma	NJAU-CVM-2018004	Da Huang	60	Gansu province	Remove accumulation with purgation, cool the blood and remove the toxic materials
Coptids rhizome	*Coptis chinensis* Franch.	Rhizoma	NJAU-CVM-2018005	Huang Lian	30	Sichuan province	Clear heat and dry damp, clear away heat and toxic material
Phellodendri chinensis cortex	*Phellodendron Chinese* Schneid.	Cortex	NJAU-CVM-2018006	Huang Bo	30	Sichuan province	Clear heat and dry damp, clear away heat and toxic material
Gardeniae fructus	*Gardenia jasminoides* Ellis	Fructus	NJAU-CVM-2018007	Zhi Zi	30	Zhejiang province	Purge fire and remove toxic substance, reduce fever and cause diuresis
Paeoniae radix alba	*Paeonia lactiflora* Pall.	Radix	NJAU-CVM-2018008	Bai shao	15	Zhejiang province	Nourish yin and supplement blood, calm the liver and relieve pain

**Table 2 vetsci-10-00328-t002:** Biochemical identification results of pathogenic bacteria.

Test or Medium	Isolate H	ATCC25922	*Escherichia coli*
TSI response mode	$\frac{A/A}{+/-}$	$\frac{A/A}{+/-}$	$\frac{A\left(K\right)/A}{+\left(-\right)/-}$
Indole	+	+	+
Methyl Red	+	+	+
VP	−	−	−
Simmons Citrate	−	−	−
Glucose	⊕	⊕	⊕
Lactose	⊕	⊕	⊕/−
Maltose	⊕	⊕	⊕
Mannitol	⊕	⊕	+
Sucrose	⊕	⊕	V
H_2_S	−	−	−
Urease	−	−	−
Power (semi-solid agar)	+	+	+/−
Phenylalanine deaminase	−	−	−
Lysine decarboxylase	+	+	V
Ornithine decarboxylase	+	+	V
Ribitol/Calendula	−	−	−/+
Inositol	−	−	−
Sorbitol	−	+	V
Melibiose	+	+	+
Raffinose	+	−	V

Note: Isolate H is the bacteria isolated from the hearts of pathogenic chickens. +, positive; −, negative; ⊕, acid and gas production; +/−, positive for most strains/negative for a few strains; −/+, negative for most strains/positive for a few strains; V, different reactions between species. Reactions on TSI (trisaccharide iron medium): $\frac{slope/bottom}{gas\;production/H2S}$; *A*, acid production (yellow); *K*, alkali production (red); ( ), occasionally visible reaction.

**Table 3 vetsci-10-00328-t003:** The antimicrobial susceptibility test and resistance parameters on the isolated strain and quality control strain.

Antibiotic	Isolate ^1^ MDREC		^2^ ATCC25922	
	The Diameter of the Inhibitory Bacterium (mm)	Determination of Drug Resistance	The Diameter of the Inhibitory Bacterium (mm)	Determination of Drug Resistance
AMP	0.0	^3^ R	18.77	^4^ S
AML	0.0	R	23.16	S
CPI	16.3	R	30.88	S
CRO	12.0	R	32.05	S
LEX	9.1	R	22.86	S
CED	10.5	R	21.85	S
GEN	12.4	R	22.90	S
AN	8.0	R	26.77	S
KAN	13.0	R	26.04	S
STR	11.5	R	18.40	S
TET	0.0	R	24.05	S
CIP	0.0	R	29.24	S
LVF	10.7	R	28.12	S
NOR	0.0	R	34.41	S
OFL	12.6	R	26.95	S
SXT	0.0	R	33.61	S
CHL	0.0	R	21.66	S
FOS	0.0	R	38.23	S
SF	0.0	R	25.64	S

^1^ MDREC: The isolated multi-drug-resistant *E. coli*; ^2^ ATCC25922: The reference strain; ^3^ R: The strain was resistant to the tested antibiotics; ^4^ S: the strain was susceptive to the tested antibiotics. Abbreviations: AMP, ampicillin; AML, amoxycillin; CPI, ciprofloxacin; CRO, ceftriaxone; LEX, cefalexin; CED, cephradine; GEN, GentamicinAN, amikacin; KAN, Kanamycin; STR, streptomycin; TET, Tetracycline; CIP, Ciprofloxacin; LVF, Levofloxacin; NOR, Norfloxacin; OFL, Ofloxacin; SXT, sulfamethoxazole and trimethoprim; CHL, chloromycetin; FOS, Fosfomycin; SF, Sulfisoxazole.

**Table 4 vetsci-10-00328-t004:** The results of MIC of TCM and CIP (unit: mg/mL).

	MDREC	ATCC25922
YJP	250.0	250.0
Rb	250.0	250.0
SR	62.5	62.5
Bac	2.0	2.0
CIP	8.0 × 10^−2^	1.0 × 10^−5^

YJP, Yujin Powder; Rb, Rhubarb; SR, Scutellariae Radix; Bac, Baicalin; CIP, ciprofloxacin.

**Table 5 vetsci-10-00328-t005:** The survival situation of each group in the experiment.

Group	Sample Number	Survival Number	Survival Rate (%)
BC	15	15	100.00 ^a^
EC	15	4	26.7 ^d^
CIP	15	5	33.3 ^cd^
YJP	15	10	66.7 ^bc^
SR	15	10	66.7 ^bc^
Bac	15	11	73.3 ^b^

BC, Blank control; EC, *E. coli* control; YJP, Yujin Powder; Rb, Rhubarb; SR, Scutellariae Radix; Bac, Baicalin; CIP, ciprofloxacin. The letter “a” indicates the maximum value, “a”, “b”, “c”, and “d” decrease in turn. The same lowercase letter indicates no significant differences between groups, while completely different lowercase letters show significant differences with a *p* < 0.05.

## Data Availability

Not applicable.

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
