# Peer review of "Bacteriostatic Effects of Yujin Powder and Its Components on Clinical Isolation of Multidrug-Resistant Avian Pathogenic Escherichia coli"

_vetsci, 2023, doi:10.3390/vetsci10050328_

Round 1
Reviewer 1 Report
I congratulate the authors for the topic and its relevance, particularly at this time of AMR concern and ONE HEALTH policy, but suggest some changes that can improve the scientific content of this paper.

Reviewer 2 Report
The author conducted a study on the bacteriostatic effects of Yujin Powder and its components on Escherichia coli. The bacteria was isolated from a sick chick in a suspected colibacillosis layer farm and identified as Escherichia coli through various methods including cell morphology, Gram staining, biochemical activity detection, and 16s rRNA sequencing. The Antimicrobial Susceptibility Testing revealed it to be MDREC. In vivo experiments were carried out on mice, which demonstrated that YJP, SR, and Bac had significant antibacterial effects by reducing bacterial loads, endotoxin release, and inflammation in vivo, as evidenced by histopathological changes in the spleen.
Critique:
While this study is relevant for controlling the APEC, there are several issues that must be addressed to increase its significance. The following are the outlined areas that require improvement.
1. The article titled "Isolation of Multidrug Resistant Avian Pathogenic Escherichia coli" needs to provide additional data to support the claim of Avian Pathogenic Escherichia coli (APEC) since the animal experiment was conducted in mice rather than chickens, the suspected host of colibacillosis. Additionally, the use of multidrug-resistant E. coli (MDREC) throughout the majority of the article instead of APEC raises questions about the accuracy of the title. Therefore, it is necessary to include more evidence to demonstrate the avian pathogenic nature of the isolated E. coli.
2. Figures 2 and 3.7 require revisions for the histopathological change of mice. Figure 2 needs to be replaced with more clear and detailed image. Additionally, the descriptions of the figures (lines 264 “the number of lymphocytes was reduced” and “Increased red blood cells” Line 265 “Many inflammatory cells infiltrated in the EC group”, Line 271 “the number of lymphocytes reduced, a large number of red blood cells and inflammatory cells were found,” Line 274 “there were still many red cells and inflammatory cells.”) need to be clarified for better understanding. It is necessary to include statistical analysis and p-values to support the findings described in these figures. Without such data, the results cannot be considered reliable. Therefore, it is crucial to include detailed statistical analyses to provide credibility to the results presented.
3. It is important to reference previous research regarding the emergence of Multidrug Resistant Avian Pathogenic Escherichia coli in the introduction. Such as, Xu, et.al, 2019 (PMID: 31572810) and Gao, et.al, 2018 (PMID: 30157463) have both reported on this issue. And these references can be used to support the urgency of developing safe and effective agents to treat multidrug-resistant E. coli. These will underscore the importance of this research.
4. The article's figure 1 E initially defines the isolation as multidrug-resistant E. coli (MDREC), creating confusion regarding the need for Antimicrobial Susceptibility Testing. It is essential to revise this aspect of the article since the presence of MDREC renders the testing unnecessary. Therefore, MDREC should be replaced by other name from the article before presenting the Antimicrobial Susceptibility Testing results. This modification will clarify the study's findings and prevent confusion among readers.
5. It is necessary to specify the number of independent experiments and the number of samples/mice used (n=?) in the figure legends for both figure 2 and figure 3. The abbreviations used in figure 3, such as "a," "b," "c," and "bc," should also be interpreted in the figure legends to enhance clarity. Additionally, the p-value is mentioned in the description of the results but not present in the figures. Therefore, it is essential to draw a line through the two groups being compared and indicate the p-value clearly.
Reviewer 3 Report
The article sent by Meng et al. on the antimicrobial power of Yujin Poder against multiresistant E. coli is of great interest to the sector given the problem that antimicrobial resistance currently represents. This study and its conclusions represent a breakthrough in the treatment of colibacillosis in poultry.
It is an investigation with a very careful methodology, although it is true that the manuscript needs some improvements:
Please correctly write the Latin names of the bacterial species on lines 12, 12, 17, 20-21, 26.
L17. In vivo and in vitro should be written in italics.
L20. Please remove the capital letter endotoxin.
L24. Please remove the capital letter from ciprofloxacin. Same in L132.
L30. "digestive tract and respiratory tract in poultry"
L42. Please specify that Yujin Powder is a herbal mix at this point.
L47-49. I recommend including Table 1 here with the Latin names of the plants that make up the Yujin Powder.
L49. Please correct the format of references [13,14].
L52. If I'm not mistaken, Qi is written with a capital letter...
L56. Please abbreviate references as follows: [15-18].
L57. Radix et Rhizoma Rhei or Rhei et Radix Rhizoma (L47)?
L87. The heart samples were supposedly collected postmortem. Did the animals die from the disease or were they euthanized for the study? If so, what method was used? How much time elapsed from the death of the individual until the sample was collected? Was it kept cold? How much time elapsed from sample collection to inoculation in nutrient broth? Was the sample kept in any specific means of transport?
L88-90. The presence of bacterial colonies in the heart sample reflects an infection in cardiac tissue that may or may not be related to diarrhea. Given the pathogenesis of colibacillosis in birds, this is to be expected, but I suggest removing this sentence so as not to confuse the reader. In addition, as it is formulated, it is not a phrase about methodology, but about results.
L126. What is SR? It is not specified in the text. The same occurs with Rb (L127), MIC (L132), and with ICR (L146).
L153. CIP was administered by hypodermic injection... subcutaneous? intradermal? intramuscular? Please specify.
L158. Was a spleen sample collected by biopsy at 18h and another at 72h? In line 160 it specifies that whole spleens were fixed and stained... Please reformulate.
L165. How long were the samples incubated? Please specify...
L197. Identification (in the singular).
L201-202. Calendula lateralis is a plant? I am familiar with Calendula officinalis. If it is a species of plant it should be in italics...
L206. Table 2: in the columns, it is clear that ATCC25922 is the standardized strain of E. coli, but it is not clear which is the strain isolated from the heart, since in the footer of the table it is named H. Please correct this mistake for a better understanding.
L228-236. Unify everything in a single paragraph just below the table.
L244. Indicate the p-value that justifies this. Same on line 252.
L246. Table 4: Please indicate in the footer the meaning of all abbreviations (MDREC, YJP, Rb, SR...). The same with Table 5, indicating what the superscripts a, b, c, and d mean.
L253 and following. Given the N that the study has, results should not be expressed with two decimal places, since in order to obtain that precision the study should have at least 1,000 samples. I recommend authors express the results with a single decimal.
L264. Please change loose to lost.
L275. Include Figure 2 here, changing HEC to MDREC. Also, write the name of Escherichia coli in italics in the title of the figure and consolidate everything (lines 286-291) in a single paragraph.
L283-284. Reviewing Figure 3, in my opinion, this is not so obvious. I recommend the authors not expose subjective opinions in this section.
L293-298. Write the name of Escherichia coli in italics in the title of the figure and unify everything in a single paragraph.
L309. Please italicize the p of the p-value.
L327-329. As this sentence is written, it seems that antibiotics are as dangerous as chemotherapy drugs. In addition, it takes away the importance of the phenomenon of resistance, which is the most important thing of all... Please reformulate.
L333-334. Chinese herbal compounds and the components of Chinese herbs are the same, remove one of the two.
L338-337. Add a reference that justifies this statement about MICs.
L339. Please write BAC as before.
L341-342. Add a reference that justifies this statement about the survival rate of the animals.
L355-356. Add a reference that justifies this statement about endotoxins.
L381-383. Add a reference that justifies this statement about the use of YJP.
Regarding the supplementary material, I recommend introducing methods 1 and 2 in the text, in their corresponding sections.
Supplementary Table 1 is titled "Number of colonies" and yet the data expressed in it are colony-forming units. Modify the title and add information about abbreviations below.
Supplementary Table 2 divides the animals into 6 groups A to E and BC. I guess BC is the blank control, but... what about the others? what do they correspond to? Please specify...
Round 2
Reviewer 1 Report
The authors presented significant improvements both in terms of methodological procedure and scientific language.
However, some of my suggestions were not taken into account, which, in my opinion, damages the quality of the paper.
Author Response
我深入研究了您提供的建议,并对我的稿件进行了相应的修改。我在给你的答复中介绍了相应的修改。如果您认为需要任何其他修改,请提供具体问题编号。我会认真对待您的宝贵建议,并尽快做出相应的修改。

Reviewer 2 Report
Minor:
1. The numerical format representing the reference citations in the text needs to be standardized. Line 355, "6, 37" has a period, while "38, 39" (line 358) has a space and a period before it.
2. Line 326 states "P < 0.05," while line 331 states "P<0.05." The format should be standardized. One has a space while the other does not, and similar inconsistencies exist.
Reviewer 3 Report
First of all, I would like to congratulate the authors for the improvements they have made to the manuscript. Despite not being able to read the response to my review (the attached document does not correspond to my report), I have detected that almost all the suggested changes have been introduced in the text.
I have just a couple of suggestions:
L171-173: "After one hour, the treatment groups were treated with YJP (3.7500 g/kg) and SR (1.0000 g/kg) by irrigation, Bac (0.0375 g/kg) by hypodermic injection, and CIP (0.0200 g/ kg) by intramuscular injection, respectively." Administration by irrigation is incomplete: is it intravenous irrigation? subcutaneous? intraperitoneal? The same occurs with hypodermic injection: hypodermic is the needle used, is the administration subcutaneous, intradermal, intramuscular, or intraperitoneal? Please detail this in the sentence. Furthermore, the dosage of the molecules should be simplified as much as possible: YJP (3.75 g/kg), SR (1 g/kg), Bac (0.0375 g/kg), and CIP (0.02 g/kg).
L358: "...treatment of human diseases. [38, 39]." Remove the dot before the references.
For my part, when these minor modifications are made, the article could be published.
Author Response
Thank you for your suggestion. There was a problem uploading files in the system before, so I sent the corresponding response to the editor's email. It may be that the reviewer's serial number in that step was confused. I apologize. Thank you again for your advice and assistance. The reply this time has been checked for several times, please see it in the attachment.
